# Application of a Protocol Based on Trap-Neuter-Return (TNR) to Manage Unowned Urban Cats on an Australian University Campus

**DOI:** 10.3390/ani8050077

**Published:** 2018-05-17

**Authors:** Helen Swarbrick, Jacquie Rand

**Affiliations:** 1Australian Pet Welfare Foundation, Kenmore, QLD 4069, Australia; jacquie@petwelfare.org.au; 2Campus Cats NSW, Kensington, NSW 2033, Australia; unswcampuscats@yahoo.com; 3School of Optometry and Vision Science, The University of New South Wales, Sydney, NSW 2052, Australia; 4School of Veterinary Science, The University of Queensland, Gatton, QLD 4343, Australia; j.rand@uq.edu.au

**Keywords:** unowned urban cat, cat management, trap-neuter-return, TNR

## Abstract

**Simple Summary:**

In Australia, management of the unowned urban cat population is a continuing challenge. This is because the numbers of cats culled in trap-and-kill programs are inadequate to balance the breeding rate of the remaining cats, and also because of immigration of sexually active cats from surrounding areas in a “vacuum” effect into areas where culling has been applied. In contrast, programs based on management of cat reproduction, such as trap-neuter-return (TNR), supplemented by rehoming of socializable cats and kittens, have been shown to produce significant reductions in free-living cat populations. However, evidence is lacking that these approaches are effective in an Australian context. In this paper, we document a nine-year TNR program on an Australian university campus, supplemented by rehoming, that reduced a free-living cat population from 69 to 15 cats, while also rehoming 19 campus-born kittens and managing a further 34 immigrant cats that either joined the resident colony (*n* = 16), were rehomed (*n* = 15), or died/disappeared (*n* = 3). Subsequent institutional support for the program was strong because of a reduction in complaints from campus staff and students, the minimal institutional costs, and the improved health status of the remaining cats, all of which are desexed, microchipped, registered and fed on a daily basis.

**Abstract:**

In August 2008, the University of New South Wales (UNSW) in Sydney, Australia, commenced a trap-neuter-return (TNR) program to manage the population of approximately 69 free-roaming unowned urban cats on its Kensington campus. The goals of the program included an ongoing audit of cats on campus, stabilization of cat numbers through TNR, and a subsequent reduction in cat numbers over time while maintaining the health of remaining campus cats. Continuation of the TNR program over nine years resulted in a current population, as of September 2017, of 15 cats, all desexed (78% reduction). Regular monitoring of the cats through a daily feeding program identified a further 34 cats that immigrated on to campus since initiation of the program; these comprised 28 adult cats (16 unsocialized, 12 socialized) and six solitary kittens. In addition, 19 kittens were born on campus, 14 of which were born to immigrant pregnant females. Unsocialized adult immigrants were absorbed into the resident campus population. Where possible, socialized adult immigrants, solitary kittens, and campus-born kittens were removed from campus through rehoming. Overall, reasons for reductions in the cat population (original residents, immigrants, campus-born kittens; *n* = 122) included rehoming or return to owner (30%), death/euthanasia (30%) and disappearance (29%). This successful animal management program received some initial funding from the university to support desexing, but was subsequently funded through donations, and continues with the university’s approval and support.

## 1. Introduction

Under Australian legislation, cat populations are often categorized as domestic (owned) or non-domestic (feral), with unowned urban cats occupying a grey area between these two categories. Unowned urban cats may be considered as domestic or non-domestic depending on individual state legislation. If classed as non-domestic they are subject to legislation relating to feral pests, and if domestic, they are subject to animal welfare legislation including laws relating to animal abandonment. Unowned urban cats are found in every urban and peri-urban area in Australia. They live close to or around human structures such as houses, factories, schools and shopping strips [1], and survive primarily through casual or regular feeding by the local community, by scavenging human rubbish and by catching prey, mainly introduced mice and rats. Such cats often congregate in colonies and thus are sometimes termed “colony cats”. Other terms often used to refer to unowned urban cats include “free-living”, “free-roaming”, “stray”, “street” or “community” cats.

Although they are of the same species (*Felis catus*), unowned urban cats are usually considered to be distinct from truly feral cats, which do not depend on humans for shelter or food, and are usually found in rural or forested areas some distance from human habitation [2,3]. The primary food source for feral cats is through predation, and thus they may impact significantly on Australian native wildlife. Management of feral cats is a significant challenge [2] and is beyond the scope of this paper.

Unowned urban cats largely comprise semi-owned or unowned domestic cats that may receive some care from persons who do not perceive themselves as owning the cat. In an Australian internet survey, 9% of respondents fed a cat daily they did not perceive that they owned [4], and 22% of households provided occasional care to an unowned cat [5]. Some apparently unowned urban cats are lost or wandering owned cats. The distinctions between unowned and owned domestic cats can become blurred and cats may move across categories in certain circumstances [4,6,7]. For example, an owned domestic cat that is lost or abandoned may join a colony of unowned urban cats. Similarly, unowned urban cats may be adopted to become an owned domestic cat. To add to the complexity, some semi-owned cats are desexed and microchipped and some owned domestic cats are not [4]. Unowned and semi-owned domestic cats constitute a large proportion of animal shelter admissions [8,9].

The approach to management of unowned urban cats varies considerably around Australia depending on state laws, council by-laws and practices, and local community attitudes. Perceived problems with unowned urban cat populations include uncontrolled breeding and population growth, nuisance behaviors such as fighting, spraying and roaming, the potential for cats to harbor diseases such as toxoplasmosis, and the perceived threat to native wildlife [3,10,11,12,13,14,15]. On the other hand, many individuals and community groups, either openly or covertly, support and feed colonies of unowned cats, often combining this with rescue and adoption of young cats and kittens, and management of reproduction through desexing [1,3,6,16]. These people are motivated by compassionate reasons, including a desire to stop cats breeding and to make their life better and safer, positive perceptions of themselves and from others for feeding a stray cat, and also the role of free-roaming cats in rodent control [1,4,6].

The principal official approach taken to manage unwanted urban cat populations is to trap and kill (cull) those colonies that give rise to persistent complaints from the public. This approach to cat population management has not been demonstrated to be effective in reducing cat numbers in the medium to long term [17,18], mainly because the number of cats that can reasonably be culled is an insufficient proportion of the population to exceed the replacement rate through breeding and immigration [10,19,20]. In particular, culling creates a “vacuum” effect that encourages immigration of other cats from surrounding areas [2,12,13,14]. This management strategy also places major stresses on shelters and municipal pounds (local government animal facilities) and veterinary clinics that must cope with euthanasia of otherwise healthy cats and kittens, not only in terms of resources, but also in emotional and ethical costs for staff involved, including post-traumatic stress which can lead to depression, substance abuse, high blood pressure and, tragically, suicide [21,22,23,24]. There is currently significant interest in Australia, and elsewhere, in reducing this burden on pounds and shelters, such as demonstrated by the activities of the G2Z organization (www.g2z.org.au), and the Australian Pet Welfare Foundation (www.petwelfare.org.au).

An alternative management strategy, which is being used increasingly in other countries such as the USA and in Europe, is trap-neuter-return (TNR) [7,12,13,14,16,17,18]. This strategy involves trapping unowned cats, desexing them, and returning them to the location where they were trapped to resume their life with their colony. This management strategy frequently also involves removing young kittens and socialized adult cats for rehoming, contributing to a rapid reduction in colony size, and this is considered best practice [1]. In TNR programs, associated animal management at the time of desexing typically includes vaccination and parasite treatment, treatment for minor health issues, and in some cases microchipping, or ear-tipping or tattooing for later visual identification.

Rarely, a cat cannot be returned to the location where it was trapped because of safety, environmental or practical reasons—in this situation, an alternative location must be found for release [25]. Cats considered too unhealthy to be returned to their home location are humanely euthanized, typically fewer than 1% of cats [26].

A number of published studies, including from the USA, UK and Italy, have demonstrated that TNR in a targeted area can be successful in reducing a cat population over time, particularly when augmented with rehoming of kittens and socialized adults. For example, Levy and co-workers [7] reported a reduction in free-roaming cat numbers on a Florida university campus from 155 to 23 over a 11-year period under a TNR program. Natoli and colleagues described a reduction in free-roaming cat numbers in Rome after a period of TNR management [16]. In a recent paper, Spehar and Wolf [17] described a TNR program conducted in Massachusetts, which reduced free-roaming cat numbers from approximately 300 to zero over a 17-year period. In a separate paper, Spehar and Wolf [18] also detailed a reduction in cat numbers in an area of Chicago managed over a 10-year period by a neighborhood TNR program. Earlier studies by Neville and Remfry [12], Neville [13] and Zaunbrecher and Smith [14] also detailed reduction or stabilization of small cat colonies using TNR. Recently, a survey of TNR practices in Australia reported that this management approach resulted in a median reduction in colony numbers from 12 to 7 cats over a median period of two years [1]. Although this study was uncontrolled and relied on self-reports of people engaged in TNR, it does suggest that this management approach may be successful in an Australian context for reducing urban stray cat numbers.

In studies involving larger populations rather than well-characterized colonies, it is challenging to enumerate the cat population accurately to document effectiveness of management approaches over time, although methods are available to monitor change in cat density in a subset of the population [27]. Changes in rates of shelter impoundment are therefore often used as a surrogate for reductions in the overall unowned cat population. Thus, when TNR was applied rigorously in a Florida zip code, shelter impoundment of unowned cats decreased by 66% over a 2-year period, representing a decrease in admissions from 13 to 4 cats per 1000 residents, compared to a 12% reduction in non-target areas [26].

On the other hand, there have been reports of TNR programs where cat numbers have not reduced or have even increased [28,29,30]. In these reports, the reasons for failure to reduce overall cat numbers included not desexing a significant proportion of the female cats [28,29], immigration of cats into the colony from surrounding areas and failure to manage these adults and resulting kittens by desexing or adoption [29], and dumping by humans of unwanted animals [30]. These effects, which counteract the impact of TNR in managing populations, have also been noted as a factor reducing the effectiveness of population control even in reports of successful TNR programs [14,16,17].

In any non-lethal cat management program, it must therefore be recognized that the TNR process acts to reduce or limit the growth rate of the colony through reproduction, but may not in itself reduce cat numbers. Any population reduction occurs primarily through adoption of socialized cats and kittens, natural death or euthanasia of sick animals, and disappearance or emigration of cats [6,7,12,13,14,16,17,18,31]. TNR works together with these factors to reduce reproduction and thus to minimize replacement of animals lost from the colony. Nevertheless, other factors such as immigration of cats from surrounding areas can counteract this effect. Thus, the impact of TNR interventions on unowned urban cat populations can be complex, and ongoing management of colonies becomes an important component in optimizing reductions in the cat population [6,7,12,28].

The purpose of this paper is to report outcomes of a carefully-managed and rigorously-applied TNR program conducted over nine years on an Australian university campus. This study provides strong evidence for the effectiveness of targeted TNR, supplemented by rehoming and management of immigrant cats, to humanely reduce the numbers of unowned urban cats in an Australian context.

### Campus Background

The University of New South Wales (UNSW) campus at Kensington is an urban university campus of approximately 38 hectares in the eastern suburbs of Sydney, which was established in 1949 on the site of previous racing stables and racecourse. Its immediate environs include a major racecourse and stables to the north (Royal Randwick racecourse), a suburban shopping strip to the south (Kingsford shops), a large hospital complex to the east (Prince of Wales hospital), and surrounding suburban residential areas (Kensington, Randwick and Kingsford). The campus currently has a human population of approximately 53,000 students and 6000 academic and support staff.

The campus has probably always carried a resident unowned cat population from its days as a horse racing property, when cats may well have been encouraged to control vermin. Since its establishment as a university, the cats on campus have been sporadically fed by students, staff and community volunteers, and occasional attempts were made to manage the increasing population through trapping and desexing. However, there was minimal involvement of university management in these activities, and no financial support. Consequently, attempts to limit and manage the university cat population had little success, and the population was largely uncontrolled.

In 2000, an informal organization called the Campus Cat Coalition (CCC) was formed to apply TNR principles for management of the cat population, with regular cat feeding by university staff. There was some consultation with university management but no formal support. Indeed, strong opposition to the presence of any cats on the campus was voiced in some parts of the university, mainly in relation to potential impacts on campus wildlife and the threat of disease to campus users. Because of a lack of university support or funding, the CCC program began to falter.

In January 2008, some kittens were found dead in the basement of a university building, concomitant with a flea infestation in that building which adversely affected the health of some university staff. A decision was made by university management that the presence of cats on the campus constituted a work health and safety (WHS) issue, and a cat trapping and eradication program was instigated by the university.

A pest management company was contracted to remove the cats. In February 2008, four campus cats were trapped over a five-day period. One cat was euthanized, one kitten was rehomed from the local council pound (local government animal control facility), and two cats were released from the council pound to CCC members for rehoming. In April 2008, over a ten-day period a further 12 cats were trapped and all were euthanized. Because it was subsequently established that some of the euthanized cats were already microchipped and desexed, and also as a result of opposition from some university staff and others to this culling operation, trapping ceased late in April 2008.

In June 2008, the CCC made a formal approach to the peak university WHS committee, proposing an alternative, more effective and humane approach to management of the campus cat population based on rigorous application of TNR principles. Discussions with university management resulted in approval, in August 2008, for a one-year trial of TNR with the following defined goals and targets:Immediate and ongoing audit of cat numbers on campusStabilize current cat numbers through a TNR program, prioritizing desexing of females, then males; 95% desexing target by the end of 2009Reduce cat numbers over time through rehoming, humane euthanasia of sick animals, and natural attrition; target 20% reduction by the end of 2009, eventual target 50% reduction in five yearsMaintain the health of the campus cat population by regular feeding, routine vaccination and parasite management, and veterinary care where necessaryEducate and inform the university community about the cat management programFacilitate ongoing oversight of the campus cat program through regular meetings with university management, and effective rapid response to emerging problemsComprehensive review of program outcomes and progress towards goals at the end of 2009.

Although some funding (approximately $9000 AUD) was provided by the university to support initial desexing and microchipping, the program was intended subsequently to become financially sustainable through donations and fundraising by the CCC.

The TNR program achieved its initial goals set with the university by the end of December 2009, and has continued with the approval of university management over subsequent years. This paper reports on the outcomes of the program over the nine-year period from its establishment at the end of August 2008 when a formal agreement was signed with university management, up until the most recent audit of cats at the end of September 2017.

## 2. Materials and Methods

### 2.1. Data Collection and Management

From the beginning of the TNR program in August 2008, an ongoing careful audit of campus cats was conducted on the basis of daily observations of cats seen at feeding sites and other campus locations, by members of the CCC and other interested members of university staff and students. Cats were given names for easier identification, and photographs and descriptions were recorded and cross-referenced amongst observers. Later in the program, outdoor battery-operated motion-activated infrared trail cameras were used to document and identify campus cats. A database of cats observed on campus was established, including names, approximate ages, descriptions of markings and behavioral characteristics, sex and desexed status, vaccination status and microchip numbers. This database was updated on a regular and ongoing basis. Input from members of the CCC who had cared for the campus cats prior to initiation of the TNR program was also incorporated. Over the course of the nine-year program, the CCC team was able to estimate the campus cat population on an ongoing basis with a high level of confidence.

For this study, details of the cats on campus commencing in August 2008 were retrieved from database records retained by the CCC, email correspondence, annual reports to the university management, and reports to Annual General Meetings of Campus Cats NSW, the registered animal welfare charity overseeing the campus cat management program. Extensive cross-referencing was employed to maximize data accuracy and to resolve inconsistencies in cat statistics where possible. Data were summarized at three-monthly and annual intervals commencing in September 2008 until September 2017.

Data were stratified in terms of sex of cats and desexed status, where possible. Records had not been kept on whether campus cats were entire or desexed before the start of the program. Much of this information was determined from anecdotal evidence, based on the memories of those who had been caring for the cats previously. In other cases, the status of campus cats was determined over time based on observations of physical characteristics and behavior (spraying, fighting, testicles for male cats; pregnancies for female cats), or when cats were trapped for desexing.

Data for ages of cats were not readily available because of the absence of records prior to the start of the TNR program. Approximate ages of cats were recorded on the CCC database based on estimates from veterinarians at the time of desexing after the program had commenced, whereas the ages of cats that were already desexed were based on anecdote and the memories of those who had participated in campus cat care prior to August 2008.

The university campus is not an isolated tract of land or an island, and it was inevitable that cats from the surrounding areas immigrated on to the campus during the course of the nine-year TNR program. Immigrant cats were typically detected by cat feeders when they appeared at cat feeding sites, and this was confirmed where possible by the use of trail cameras. In some cases, particularly when young kittens were found, these were reported to the CCC directly by members of the university staff who had noticed their presence.

Data on immigrant cats that were discovered on campus during the program were gathered with as much pertinent information as possible about age, sex and desexed status. These immigrant cats were categorized as either unsocialized (based on behavioral characteristics such as extreme wariness, avoiding interaction with humans, hissing and aggressive vocalizations when approached) or socialized (approachable and able to be handled). Solitary kittens that were discovered on campus were placed in this latter category, whereas kittens born on campus were recorded separately.

Unsocialized immigrant cats were absorbed permanently into the campus cat population (and were targeted for desexing), whereas socialized immigrant cats and kittens, and kittens born on campus, typically remained only temporarily on campus before transfer to other agencies for rehoming. Unsocialized (permanent) immigrants, socialized (temporary) immigrants, and kittens born on campus, were reviewed separately in this paper. Because unsocialized cats were absorbed into the campus cat population, global summaries of resident cat numbers over time included these cats as part of the population under ongoing management on the campus.

### 2.2. Desexing, Feeding, Monitoring and Health Management

Cats undergoing TNR were desexed at two local private veterinary clinics. At the time of desexing, all cats were vaccinated, treated for parasites, and received minor health care if needed. Identification was by microchipping and registration on a state database through the local government agency, as mandated by state law. Cats were registered as owned by Campus Cats NSW, and one of the office-bearers of the organization was named as primary carer.

Daily feeding of all cats on campus was achieved through the establishment of a cat feeding roster that involved volunteer students and staff of the university and some local community volunteers associated with animal welfare agencies. Cats were routinely fed once a day at around dusk at a number of locations around the campus, discretely hidden from passersby in gardens or behind hedges, and behind or under buildings or external staircases. Cats fed at these sites ranged from approximately one to eight cats per site. Both wet canned food and dry kibble were provided, and some feeders occasionally included raw meat or raw chicken wings as a supplement. Fresh water was also always available at all feeding sites and replenished daily. Some feeding stations incorporated small feeding shelters to protect food from the elements and scavenging birds.

As well as helping to maintain the health of the cats, the regular feeding routine provided an opportunity for volunteers to check cats for signs of ill health, identify new (immigrant) cats that appeared on campus, and alert CCC management if any cats were not seen for a few days so that searches for missing cats could be instigated. Further monitoring of cats was achieved later in the program through the occasional use of motion-detection trail cameras placed at feeding stations, and these were particularly useful to locate apparently missing cats that had relocated to different sites, and to monitor the return of cats to their usual feeding sites after TNR or veterinary treatment.

Where indicated, veterinary care was provided for cats. In some cases, this meant that cats needed to be trapped for transport to the veterinary practice. Because of the time required to trap sometimes trap-shy individuals, and because of the stress this could cause these cats, the initiation of veterinary care was only considered once other options, such as on-site provision of medication under veterinary supervision, had been attempted. In more recent years, as cats became more socialized to their feeders, it became increasingly possible to place certain cats in carry cages for transport to the veterinary practice and, in some cases, for full veterinary examination to be achieved in unsedated campus cats. Usually, however, sedation was required to allow full examination.

## 3. Results

### 3.1. Audit of Campus Cat Numbers over Time

At the start of the program in August 2008, the number of cats on campus was uncertain and uncontrolled breeding made this a moving target. Records had not been kept previously on cat numbers, sex, description, age, or whether cats had been desexed or microchipped. The initial CCC estimate was between 75 and 90 campus cats at the time of preliminary negotiations with university management, but this was subsequently modified as numbers became clearer through an intensive ongoing audit of cats around the university campus by CCC members and other interested campus users. In particular, cats were observed and counted during daily feedings, which were conducted at a regular time of day to encourage cats to come to the feeding stations. On the basis of these careful observations, a total of 74 cats was thus recorded in August 2008.

Once the CCC became more familiar with, and confident about, the campus cat population through the careful ongoing audit process, it became apparent that five cats (5/74; 7% of resident population) either had been double-counted (*n* = 3) or had been local wandering domestic cats (*n* = 2) at the time of the initial August 2008 population audit. The “overcount” reflects early uncertainty about cat numbers, and similar issues are likely to occur early in any TNR program. As a result, the overall baseline population count was adjusted to 69 cats in CCC records at December 2008, and for all summaries in this paper a baseline population of 69 cats was used.

In addition to the original 69 cats present on campus at the beginning of the TNR program, a total of 34 immigrant cats or solitary kittens were identified over the nine-year program, and 19 kittens were born on campus to resident or immigrant cats.

#### 3.1.1. Unsocialized Immigrant Cats

Of the 34 immigrant cats, 16 cats (47%) were categorized as unsocialized based on their behavior (wariness, fear of humans, aggression when approached) and were deemed unsuitable for rehoming. These cats were consequently managed through TNR and absorbed into the campus cat resident population. Thus, the total resident cat population managed on the campus during the nine-year TNR program was 85 cats (69 originals + 16 unsocialized immigrants) (Figure 1 and Appendix A).

#### 3.1.2. Socialized Immigrant Cats and Kittens

In addition to 16 unsocialized immigrant cats absorbed permanently into the campus cat population, a further 18 socialized immigrant adult cats or solitary kittens were present temporarily on the campus for short periods, usually between one day and two weeks (mode 1–3 days), before their permanent removal, typically for fostering and adoption.

Socialized immigrant adult cats (*n* = 12) were thought to either be lost or wandering owned or semi-owned cats, or have been abandoned by students or nearby residents, and were identified as socialized by their approachability and friendliness to humans.

Six immigrant kittens simply appeared on campus, including two kittens found in the engine bay of a student’s car parked on the campus, three kittens found wandering by themselves on campus, and one injured kitten brought on to campus by a student. Solitary kittens were typically fostered and rehomed as soon as possible after their discovery.

#### 3.1.3. Kittens Born on Campus

A total of 19 kittens were born on campus during the nine-year program. Five of these kittens were born to two resident cats—the last resident kitten was born in December 2009. Both resident mothers were trapped and desexed.

In addition, a total of 14 kittens were born to three newly arrived immigrant female cats. A litter of six kittens was discovered in March 2009 with a friendly adult female. In December 2014, an unsocialized immigrant female cat (seen occasionally over the previous two months) had a litter of four kittens, and again in January 2017 an unsocialized immigrant female cat (noted sporadically over the previous 2–3 months) was found with a litter of four kittens. These 14 kittens were fostered and rehomed. The socialized mother was rehomed, and both unsocialized mothers were desexed and returned to campus.

### 3.2. Desexing of Campus Cats

At the beginning of the program (August 2008), 10 (14%) of the 69 resident cats on campus were of undetermined sex (Figure 1 and Appendix A). Sex and desexed status were subsequently clarified at the time of TNR, except for one original resident that disappeared before its status could be confirmed. Based on current knowledge of the sex and desexed status of resident cats at the start of the program, there were 33 (49%) males and 35 (51%) females, of which 48% of males (16/33) and 26% of females (9/35) were entire (Figure 2).

Socialized immigrant cats and kittens were not included in this summary of sexual status of campus cats, because these cats and kittens were rapidly removed from the campus and thus did not form part of the permanent resident population, and also because the majority of the kittens were of uncertain or unrecorded sex. Cats whose status remained unknown (*n* = 4 through the nine-year program), primarily because they disappeared before they could be trapped, were also not included.

Immediately prior to, and on commencing the agreed management program in August 2008, the CCC targeted fertile females for desexing (Figure 2A). By December 2009, all but one of the 26 female cats remaining on campus had been desexed—this last fertile female proved elusive but was finally trapped and desexed in December 2010. Since then, all resident female cats on campus were desexed, and no kittens have been born to resident cats since December 2009.

Male cats were also trapped and desexed from the beginning of the program, although this had a lower priority than female cats. By December 2009, 16 of the remaining 22 male cats (73%) were desexed (Figure 2B). At the most recent audit, all of the remaining male cats on campus were desexed, although the status of one long-term immigrant visitor (assumed male and desexed) has not been confirmed.

Unsocialized immigrant cats that were absorbed into the program after 2008 were immediately targeted for TNR and 13 of these 16 cats were successfully desexed (four females, nine males). One male cat arrived on campus with an ear tip, and was assumed to have been previously desexed. Another undesexed male disappeared before he could be trapped for desexing, and one unsocialized immigrant visitor remains of unconfirmed status.

In terms of desexing, socialized adult immigrant cats (*n* = 12) were dealt with on a case-by-case basis. These cats were initially scanned for a microchip, and this resulted in the return of three cats (25%) to their owners. The owners of two other microchipped cats were not able to be contacted, or did not want the cat returned. Cats were then checked by a veterinary surgeon for general health, and where indicated were desexed (*n* = 2). All cats received vaccinations and parasite treatment, before placement in foster care prior to rehoming. All adult socialized cats managed by this route (*n* = 7) were successfully placed in adoptive homes. Two of the 12 socialized adult immigrant cats disappeared before their status could be verified, and may in fact have been wandering local domestic owned cats that returned to their nearby homes.

### 3.3. Health Management

Health of cats was monitored by the cat feeders, facilitated by daily feeding and, where necessary, motion-detection trail cameras at feeding stations. The most common reasons for veterinary treatment among the campus cat population were eye infections, minor injuries, abscesses, skin conditions and dental disease. Various medications in oral form were successfully administered in food usually as crushed tablets, including antibiotics, steroids and other treatments such as medication for hyperthyroidism.

### 3.4. Colony Size over Time

By December 2009, when the program was due for initial review, the number of cats on campus had reduced from 69 to 51 cats, which represented a 26% (*n* = 18/69) reduction, meeting the 20% target agreed with the university. At the 5-year time point (September 2013), there were 30 cats remaining on campus, representing a 57% (*n* = 39/69) reduction, again meeting the agreed 50% target. At the time of preparing this paper (September 2017), 15 cats remain on campus, representing a 78% (*n* = 54/69) reduction in the resident cat population since the TNR program was initiated in August 2008. Case studies of some of the campus cats are included in Appendix B.

Nine of the remaining cats at September 2017 were present on campus at the start of the program in August 2008, and six were cats that immigrated on to the campus during the program. Four of these latter cats were younger than those that remained from the original cohort, most of which were now quite elderly (estimated age of original resident cats >10 years). However, two of the remaining six unsocialized immigrant cats were estimated to be older than 10 years when they first appeared on campus.

### 3.5. Fates of Campus Cats

An overall summary of the fates of all cats managed in the program over the period August 2008 to September 2017 is presented in Table 1.

#### 3.5.1. Fates of Resident Cats

Figure 3 summarizes the fates of cats from the resident (original and unsocialized immigrant) campus cat population during the nine-year TNR program.

Because many resident socialized cats on the campus had already been rehomed prior to August 2008 (in some cases to protect them from the university’s culling program), few resident campus cats were successfully rehomed after the start of the TNR program. Over the nine-year program, seven cats (7/85; 8% of resident population) were adopted, including two older cats (estimated 15 years old) that were placed in permanent homes in mid-2017. These latter adoptions became possible because of socialization of these cats through their regular daily interactions with volunteer feeders.

A total of 20 cats (20/85; 24% of resident population) were humanely euthanized during the nine-year program (mean 2.2 resident cats per year). The primary reason for euthanasia was severe ill health (*n* = 16/20, 80% of euthanized cats; Table 2).

Another 14 cats (14/85; 16% of resident population; mean 1.6 resident cats per year) died or were found dead during the nine-year program (Table 3). Five of these cats met with accidental deaths, of which three were hit by cars on roads surrounding the campus.

Based on our data on deaths and euthanasia in the resident cat population (*n* = 34/85), and the approximate number of cat-years represented in this dataset (*n* = 419 cat-years), we estimate an average mortality rate in our campus cat population (including adults and kittens) of approximately 8.1% per annum over the nine-year program.

A total of 29 cats (29/85; 34% of resident population; mean 3.2 animals per year) simply disappeared and their fates are unknown. Although some of these cats appeared to be ill or injured prior to their disappearances (*n* = 4), most appeared in good health at the time they disappeared and the reasons for their disappearances remain speculative.

#### 3.5.2. Fates of Socialized Immigrant Cats, and Kittens Born on Campus

Most (70%; 26/37) of the socialized immigrant adult cats (7/12), solitary kittens (5/6), and kittens born on campus (14/19) were managed rapidly through fostering followed by rehoming (Figure 4).

Five of the seven rehomed adults were removed from the campus on the same day they were discovered, one cat took a week to capture for foster care, and another cat was rehomed after almost three months on campus. A further three adult domestic cats with microchips were reunited on the day of their discovery with their owners, all of whom lived within half a kilometer of the university campus. Two socialized immigrant adult cats were briefly seen on campus on several occasions over a 3–5 months period, then disappeared.

Of the six solitary immigrant kittens, five were removed and placed in adoptive homes. Three were removed within three days of their discovery, whereas two took up to two weeks to capture before removal for rehoming. One kitten died within three days from severe trauma sustained prior to being brought on to campus by a concerned student.

Of the 19 kittens born on campus, five kittens were born to resident cats early in the program. Four of these kittens disappeared and are thought to have died, and one kitten was euthanized due to inoperable hernia.

The other 14 campus-born kittens (all of which were born to immigrant pregnant cats) were successfully placed in adoptive homes. Where necessary, kittens remained with their mothers on campus until weaned, and then were separated at about 4–6 weeks of age for socialization in foster care. Kittens were then transferred to a rescue/adoption agency either prior to or immediately after early age desexing (at approximately 1 kg in weight), for further socialization and rehoming.

## 4. Discussion

This study reports in detail the results obtained over a nine-year cat management program conducted on the UNSW campus using a TNR protocol to manage cat numbers, combined with adoption of socialized cats and kittens, natural attrition, and active management of immigrant cats. Over the nine-year period, 122 cats in total were managed as part of the program. These comprised 69 original residents (57%), 16 unsocialized immigrants incorporated into the resident population (13%), 18 socialized immigrant adults and kittens (15%), and 19 kittens born on campus (16%). Socialized immigrant adults and kittens, and kittens born on campus, were targeted for rapid removal for rehoming or return to their owners. This left 70% of the total 122 cats (69 original residents and 16 unsocialized immigrants; total 85 cats) to be actively managed on an ongoing basis on the campus.

The current campus cat population (at September 2017) comprises 15 cats, representing a 78% reduction in resident cat numbers from the original population of 69 cats in August 2008. Therefore, only 12% of the 122 cats managed through the course of the nine-year program remain on campus.

### 4.1. Reduction in Cat Population over Time

For the 122 cats managed during the program, the overall reduction in cat numbers over nine years resulted from rehoming/return to owner (36; 30%), euthanasia (21; 17%), death (15; 12%), and unexplained disappearance (35; 29%) (Table 1). Although comparisons of these statistics with other studies of TNR in similar populations are difficult because of differences in duration and methodology of study, and ambiguous data recording, an attempt to compare outcome data is presented in Table 4.

#### 4.1.1. Rehoming

The rehoming rate of 8% (*n* = 7) for the resident cat population of 85 in this program (Table 1) was low compared to previous studies of TNR [7,18,31]. Many adult socialized resident cats had been adopted into permanent homes prior to initiation of the TNR program, and this early phase of adoption is not reflected in our numbers. In comparison, Levy et al. reported that 47% of cats in their TNR program were rehomed [7]. This may reflect the much higher proportion in their population of kittens and younger cats, which typically form the majority of cats rehomed early in a TNR program, compared to our older campus population in which most of the resident female cats were already desexed. However, in our study, 83% of socialized immigrant cats and kittens, and 74% of kittens born on campus, were successfully rehomed, including three cats returned to their owner. When the overall rehoming/return-to-owner rate in our program is considered (36/122; 30%), this is comparable to other TNR programs that have documented rehoming rates from 21% to 32% [18,31] (Table 4).

Approximately 8% of all immigrant cats (25% of socialized immigrants) were returned to their owners who lived within 500 m of the campus. Owner contact details from microchip registration were used to locate the owners. This wandering behavior of owned cats is consistent with GPS tracking data for South Australian pet cats, where the average daytime range was 1 hectare, and 2 hectares at night [32]. It is also consistent with research demonstrating that 75% of lost cats when found were located within 500 m of their home [33].

#### 4.1.2. Euthanasia and Death

The overall euthanasia rate amongst the resident cat population was 24% (20/85) over a nine-year period, or 17% (21/122) for the total cat population (Table 1). These cases were primarily due to severe and terminal illnesses in the original adult resident cat population (Table 2). There was a higher rate of euthanasia earlier in the program. Overall, one-third of cases of euthanasia occurred during the first 18 months of the program, primarily involving younger cats. Later cases of euthanasia related to terminal but relatively common cat diseases such as renal disease and neoplasia, involving older cats.

The overall euthanasia rate is higher than Levy’s study [7] of 11% over 11 years, but differences may reflect that Levy’s study population comprised younger cats compared to our older campus cat population. Our euthanasia rate is also significantly higher than the 3% reported by Spehar and Wolf [18], but comparable to other reports [31]. This variability in euthanasia rates may reflect differences in program duration, and methodological and demographic differences between studies. It could also be argued that the close monitoring of our campus population meant that sick cats were identified quickly and euthanized following terminal diagnosis, rather than disappearing or being found dead later due to their illness. Furthermore, in most calculations of overall euthanasia rate, rehomed cats are assumed to remain alive, further confounding comparisons between studies with different rehoming rates.

Our philosophy relating to euthanasia developed during the course of the program as a result of early experiences. For example, early in the program, some younger cats were euthanized because they were found to be FIV-positive, but this practice subsequently ceased as the CCC became more knowledgeable about this condition. Specifically, the risk of false positive test results, the high cost of testing, and the low incidence and late occurrence of clinical disease were felt to argue against testing and euthanasia. Furthermore, advice from other organizations such as Alley Cat Allies [34] suggested that a better way to limit FIV in the campus population was through desexing of male cats to reduce fighting and bite wounds, the main route of transmission of this virus.

A further 16% of the resident cats, or 12% of the overall cat population in our program, died or were found dead (Table 1 and Table 3). Reported percentages of cats found dead in other TNR studies vary from 4% to 16% (Table 4), but differences in study duration and methodology make comparisons difficult. Rehoming of cats may also confound these estimates as these cats are effectively lost to follow up and are assumed to remain alive in calculations of overall death rates.

The average mortality rate based on deaths and euthanasia in our campus cat population (including adults and kittens) of approximately 8.1% per annum is consistent with the calculated death rate for pet cats of 7% per annum, based on the reported average age at death of 14 years [35]. Our rates of euthanasia and death are also comparable with those that have been reported in pet cat populations; for example, the estimated annual pet cat mortality rate was 8.3% based on a telephone survey of North American cat owners [36].

The main causes for death or euthanasia in our population, where this could be determined with reasonable confidence, included trauma, neoplasia and renal disease—the latter two causes were predominant in cats over five years of age (Table 2 and Table 3). Because of the campus-based rather than veterinary practice-based nature of this program, it is not surprising that some deaths were of unknown cause or for unspecified reasons.

Studies in several countries have reported the most common diseases leading to death or euthanasia in pet cats. Cause of death/euthanasia in cats younger than five years are predominantly due to trauma or viral and respiratory disease, whereas death/euthanasia in older cats, such as our cohort, are mainly related to renal disease, neoplasia, cardiovascular disorders, or non-specific illness [37,38]. Thus, our findings of rate and causes of cat mortality in our unowned urban cat population managed with TNR differ little from those commonly cited in relation to owned cat populations. This is consistent with previous findings that TNR programs may be associated with comparable or even increased longevity compared with owned cats [39], and that unowned urban cats in TNR programs show similar levels of health and profiles of disease as owned pet cats [40,41].

#### 4.1.3. Disappearance

The most common cause for loss of cats in our study population of 122 cats was unexplained disappearance, accounting for 29% of cats and representing a mean of 3.9 disappearances per year across the overall population (Table 1). Reported disappearance rates from similar TNR studies range from 11% to 34% (Table 4), but as discussed previously, these comparisons are confounded by differences in study duration and methodology [6,7,14,18,31]. Furthermore, these statistics assume that adopted cats do not disappear after rehoming. However, 15% of pet cat owners lose their cat at least once in a 5-year period, and 30% of pet cats go missing at least once in their lifetime [42].

The fates of cats in our program that disappeared are unknown. Likely reasons include relocation (emigration), adoption, or death (although the majority were apparently healthy at the time of disappearance). Because some cats were becoming increasingly socialized due to their regular interactions with campus cat feeding volunteers, it is certainly possible that some of these more socialized cats were simply adopted informally by students, staff or campus visitors without the knowledge of the CCC team. Others may have left the campus to join nearby urban cat colonies at the racing stables or local shopping strip, possibly in response to competitive behavior from fellow campus cats, or because they were undesexed males looking for mates.

In our study, the total number of cats that disappeared (or emigrated, 35) almost exactly balanced the number of cats that appeared (immigrated, 34). This is in contrast to the findings from a one-year Israeli study that compared numbers in desexed and entire colonies of unowned urban cats. In that study, despite high desexing rates for females in the desexed colonies, overall numbers increased compared to the entire cat colonies. Sexually intact cats more readily immigrated into the desexed colonies, whereas desexed cats were less likely to emigrate [29].

#### 4.1.4. Summary of Reasons for Reduction in Cat Population

In summary, in common with other studies of TNR in targeted populations, the main reasons for reductions in the campus cat population in our program were rehoming, death, euthanasia and disappearance. Although concerns relating to the welfare of released cats are often cited by opponents of TNR programs, the death, euthanasia and disappearance rates in our campus cat population were similar to those reported for pet cats.

Active identification and management of immigrant cats also played an essential role in control of the campus population. Immigrant cats (*n* = 34) represented a potential increase of 50% on the original resident population of 69 cats during the nine-year program and, if not appropriately managed, could have contributed significantly to population growth by approximately 4% per year. The birth of kittens to immigrant cats further increased the potential for population growth. At the same time, it must be acknowledged that our program reduced overall cat numbers (see Figure 1 and Appendix A), potentially creating the circumstances in which a “vacuum effect” may have encouraged the influx of immigrant cats. We cannot know what would have happened had the resident population remained stable and resources to support the cat population had remained constant. Nevertheless, failure of some TNR programs to control cat populations because of uncontrolled immigration of sexually intact animals from surrounding areas, and the dumping of unwanted cats or kittens, has been reported by others [29,30]. Furthermore, our program did not have the capacity to address management of the source of immigrant cats through community desexing programs for owned cats and promotion of responsible pet ownership in the areas surrounding the campus. The potential impact of such strategies on immigration of owned cats on to the campus is thus unclear.

Nevertheless, a critical factor responsible for the gradual reduction in resident cat numbers during the nine-year program was the rigorous application of desexing to limit cat reproduction, which otherwise may have maintained or even increased the cat population. Cats have a large reproductive potential. A single queen may produce an average of ten kittens per year, and female kittens can become pregnant at four months of age [43,44]. Failure to desex all (or nearly all) female cats in a colony is one of the reasons that some TNR programs have failed to reduce cat numbers [28,29]. In our campus population, 95% of females were desexed within the first 15 months of the program, and no kittens were born to resident cats thereafter. However, despite this early application of TNR, three pregnant females immigrated on to campus and gave birth to kittens, highlighting the ongoing need to monitor immigrants, manage new kittens born, and desex immigrant cats as quickly as possible.

Without active management of immigrant cats, combined with effective suppression of reproduction through TNR, it is likely that the campus cat population would have increased, as was apparent before initiation of the TNR program in August 2008. This emphasizes our view that effective TNR management cannot occur as a “one-off” intervention, but necessitates ongoing management of colonies to minimize the impacts of immigration. It should also be noted that ongoing management is also essential when culling is used to reduce cat populations, because the birth of new kittens and immigration into the culling area due to a “vacuum” effect will act to increase the cat population [2,12,13,14].

### 4.2. Health Management and Maintenance

One of the key elements of the university-approved TNR trial reported here was the maintenance of health of the cat population through daily feeding and regular monitoring of cat welfare and health status. Under the original university agreement, veterinary care was also to be provided where this was indicated.

Best practice for daily feeding of unowned urban cats usually involves the removal of any left-over food after about 30 min, to reduce attraction of vermin and also to minimize complaints [1]. Because we relied heavily on student volunteers to place food for our resident cats, this approach was not logistically feasible because of students’ other commitments. Furthermore, because of the high volume of campus foot traffic at the end of each day as students left classes, cats frequently remained hidden until later in the evening, after student volunteer feeders had left campus themselves. Our approach therefore was to prescribe carefully the amount of food left per cat, and in recent years premeasured bags of dry kibble were supplied for student feeders to emphasize limitations on the amounts of food provided per cat.

The most common reasons for veterinary treatment among the campus cat population such as dental disease, eye infections, minor injuries, abscesses, and skin conditions, differed little from the spectrum of minor illnesses seen in the domestic cat population. In our experience, tablet medications including antibiotics, steroids and even treatment for hyperthyroidism (maintained over a 12-month period) could be successfully administered crushed in the food of the affected cats. Various strategies to ensure that medications were targeted to the cats receiving treatment were adopted, including staged and separate feeding to prevent other cats in the colony from accessing the medication.

### 4.3. Arguments for and against TNR Management of Cats on Campus

The two major arguments against maintaining a population of cats on the campus are the potential threat to campus wildlife, and the potential spread of disease from cats to humans.

The UNSW campus is an urban campus surrounded by residential housing, other institutions, a shopping precinct and a racecourse. The campus environment could be described as very highly disturbed, with many buildings, paved and concreted areas to expedite student traffic, some large spaces of lawn for recreation, and some garden plantings. The wildlife on the campus consists of some native and introduced bird life, mammals such as rodents (in particular non-native rats), possums and fruit bats, and some populations of small lizards and frogs, although we are not aware of any census of campus wildlife. It is beyond the scope of this study to estimate with any certainty the potential impact of the campus cats on these animal populations. Nevertheless, an argument can be made that by reducing the resident cat population through TNR, and through regular feeding of resident cats, the impact of predation by campus cats is likely to be substantially mitigated.

Although it is clear that feral cat populations have impacted on native wildlife in rural and wild areas of Australia [15], there is no published study that establishes that cats have a significant negative impact on native wildlife populations in an Australian urban environment. A study in Perth, Western Australia [45], found that medium-sized native mammals such as possums and bandicoots were not affected by the unregulated presence of cats compared to areas where cats were prohibited or kept indoors at night. The small mardo *Antechinus flavipes*, which is considered to be highly susceptible to cat predation, was actually found in higher numbers in areas where cats were unregulated. Another Perth study [46] concluded that urbanization with associated habitat degradation had a significant negative influence on passerine bird populations, whereas there was no measurable effect of cats. In a study of 24 forest patches in the Sydney metropolitan area, black rats, possums and other birds were found to be the main predators of bird nests, and nest predation reduced when cats were present [47]. In an Australian study, prey items caught by pet cats were, in decreasing order, mice, rats, small lizards, and then common species of birds [48]. These studies indicate that our population of campus cats probably had minimal effect on the population of native campus wildlife.

Opponents of TNR typically raise concerns about released cats killing wildlife, but there is little recognition or acknowledgement that killing healthy cats and kittens damages the health of the humans involved. Research in Australia and elsewhere has found that many shelter staff directly involved with euthanasia of animals develop perpetration-induced post-traumatic stress, and other shelter staff frequently develop compassion fatigue [22,23,24,49]. These conditions damage mental health and can lead to depression and increase the risk of suicide. In the US, the suicide rate for animal rescue workers is amongst the highest in American workers, with a rate equal to firefighters and police officers [50]. These adverse effects on human mental health deserve to be considered when choosing urban cat management strategies, because traditional complaint-based programs involving trapping and killing cats, as occurred initially on the UNSW campus, result in more cats and kittens entering shelters and their resultant euthanasia compared to TNR programs [25].

A number of diseases are spread from cats to humans, but most are spread by direct contact or by fleas, and therefore pet cats pose a greater risk to humans [51]. Toxoplasmosis is known to be carried by some cats, and can cause disease in humans, wildlife and cats, but is not spread by direct contact [52]. There is evidence that, compared to trap and cull management programs, TNR may reduce environmental contamination by toxoplasmosis oocysts, thus mitigating risks of transmission to other species. In trap and cull programs, new susceptible kittens are in higher proportions due to unchecked reproduction, and these immunologically naïve kittens are more likely to become infected by the Toxoplasma parasite and to shed oocysts in the environment [53,54]. On the other hand, TNR programs, which achieve high desexing rates and actively rehome young kittens, result over time in a population of more mature cats that have a higher probability of being immune to toxoplasmosis and are not shedding oocysts. Furthermore, adult cats that are not yet immune shed fewer cysts than younger cats when infected with Toxoplasma [52,54]. The reduction in overall numbers of free-living cats through TNR also minimizes risks of Toxoplasma infection [55].

Nevertheless, the CCC was conscious of the potential for disease transmission to our volunteer feeders. Although infection occurs through ingestion of oocysts which only become infectious several days after being shed in cat feces, to mitigate the potential risk of toxoplasmosis infection during management of campus cats we recommended the use of plastic gloves when feeding or handling the cats, and emphasized the importance of personal hygiene after engaging in feeding activities. We also discussed this potential risk with university management, so that they could appropriately inform and instruct ground staff who may work in areas frequented by the cats.

### 4.4. Financial Support

An estimate of the annual costs associated with this TNR program is not possible because records of expenditure were not kept until later years of the program, and because a majority of the costs associated with the program, in particular food, were met by informal out-of-pocket contributions from feeders and other members of the CCC. Food for the campus cats was supplied as a donation by those feeders who were in full-time or part-time employment, whereas food was supplied on a complimentary basis by the CCC to enrolled university student feeders, from a central campus location. This latter food was funded through donations.

Similarly, all veterinary care including desexing, examination and treatment of ill cats, and costs of euthanasia, was funded through donations, supplemented by generous discounts from local veterinary practices. Early in the program, most veterinary costs were related to desexing, whereas later in the program most of these costs were associated with health maintenance and management of an aging cat population. In the event of substantial individual veterinary care bills, email appeals to CCC supporters were usually successful in raising the necessary funding to cover costs. Although the university initially provided some limited funds (approximately $9000 AUD) for cat desexing at the start of the program, the CCC was subsequently self-supporting as a registered animal welfare charity since 2009, through regular donations from supporters and fundraising activities.

### 4.5. Institutional Attitudes

University management was initially skeptical about the potential of a TNR program to successfully manage and reduce cat numbers, but the rapid reduction in the resident cat population and the significantly reduced level of complaints from the campus community, with no formal complaints recorded after 2009, quickly changed institutional attitudes towards the program. This was enhanced by eliminating costs sustained by the university in managing the resident cats, such as costs associated with previous culling of cats, and with complaint management and resolution. Furthermore, the ongoing direct costs relating to feeding, veterinary care and desexing were borne by the animal welfare charity and volunteers.

Regular meetings between CCC and university management expedited increased cooperation. There was a noticeable difference in practical support from university divisions such as facilities/estate management, security and grounds/landscaping teams, as a result of the clear and apparent improvement in the management of the campus cat population, the improved health of the resident cats, and the manifest reduction in numbers.

These advantages of TNR programs are also likely to be apparent in community-based TNR programs to manage unowned urban cats. Much of the cost to local councils and governments of trapping and holding cats for minimum periods mandated by Australian state legislation, prior to adoption or killing, can be minimized not only through reduction in the unowned cat population through TNR, but also because of volunteer support and input from local community groups [1]. Furthermore, based on our campus experience, a significantly reduced level of complaints and enhanced local community support for more humane management strategies for unowned urban cats are likely to result from the application of TNR principles.

### 4.6. Limitations

This study has a number of limitations, as would be expected from a retrospective study. The TNR program on the university campus was not initiated as a formal scientific study, but to provide humane management of a free-living unowned urban cat population. At the outset of the program, there was no intention that program outcomes might be formally reported in this paper. No scientific protocol was developed before commencing the program in terms of cat management, nor was there a predetermined schedule of checking or trapping the cats. Management of the cats was developed informally by a small group of untrained cat lovers and citizen scientists, rather than by veterinary scientists.

Although the enumeration of cats on the campus during the nine-year program was carried out with diligence and care, it is possible that some cats, particularly if present only transiently on campus, may not have been recorded. No clear data on age of cats was available, because of the absence of records before initiation of the TNR program in August 2008. Records of costs of the TNR program were not maintained, and in any case many of the costs were “hidden”, having been met informally by donations from members of the CCC and other supporters of the program. In-kind contributions, in particular discounts from veterinary practices, were also not readily quantifiable. These factors precluded comparisons with the costs that might have been incurred if a trap and cull approach, as initially pursued by the university, had been continued.

The applicability of our findings outside the limited and defined range of a university campus or similar site is unclear. Immigration might constitute a larger component where colonies are immediately adjacent to houses with free-roaming owned cats, given the limited range of most pet cats [32]. Furthermore, in common with many previous reports of TNR programs, no control group was included to document and compare changes in cat numbers in an unmanaged population. Further prospective research is therefore recommended to confirm outcomes of this TNR program.

## 5. Conclusions

In summary, a nine-year cat management program conducted on the UNSW Kensington university campus in Sydney, Australia, successfully reduced the campus cat population through a combination of TNR and rehoming. At the beginning of the program, there were 69 resident cats. A further 34 immigrant cats (16 unsocialized and 18 socialized), and 19 kittens born on campus, were documented on the campus during the program. At the time of reporting (end of September 2017), the campus has a resident population of 15 cats. All current resident cats are desexed, clinically healthy and managed on a daily basis by volunteer feeders. Many of these cats are now elderly (>10 years of age), and it is anticipated that resident cat numbers will decline over the next few years due to natural attrition. At the same time, an inevitable continuing influx of immigrant cats is expected to supplement current population levels, necessitating continuing active management through ongoing TNR and rehoming.

## Figures and Tables

**Figure 1 animals-08-00077-f001:**
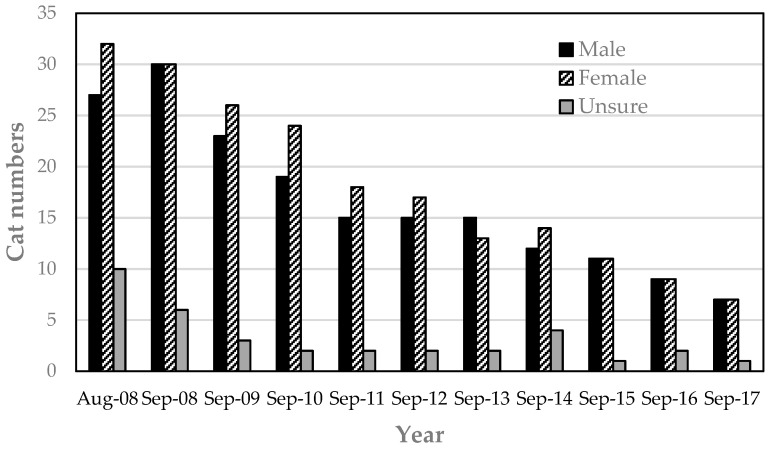
Resident campus male and female cats (including 69 original cats and 16 unsocialized immigrants) at commencement of the program (August 2008) and at annual intervals from end of September 2008 to end of September 2017.

**Figure 2 animals-08-00077-f002:**
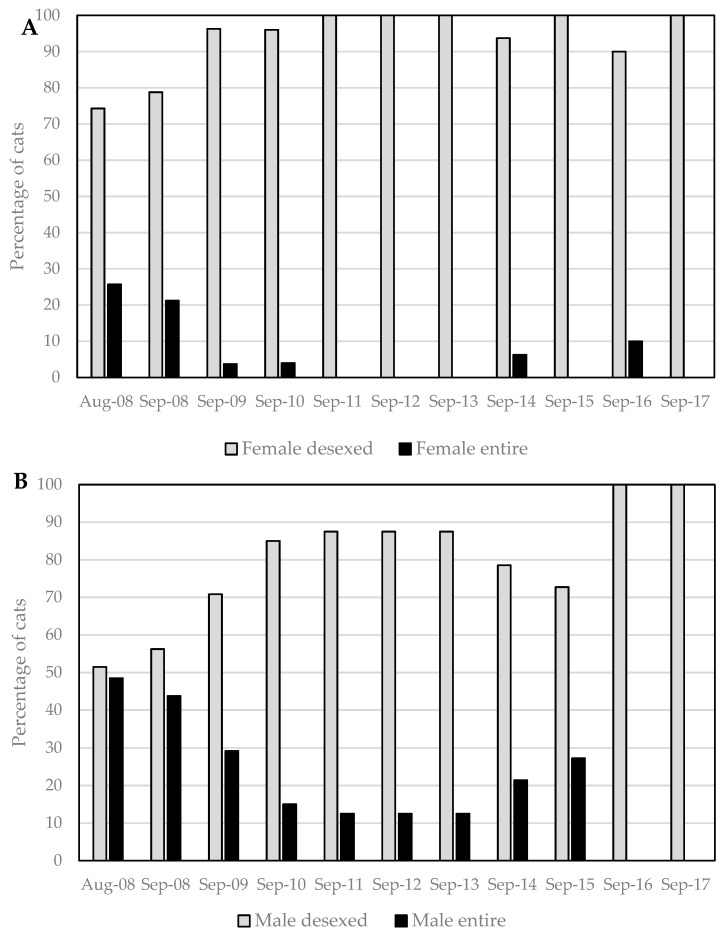
Percentage of resident cats (including 69 originals and 16 unsocialized immigrants) desexed versus entire (undesexed) at annual intervals during the nine-year program—(**A**) female; (**B**) male. Cats of unknown status (*n* = 4) were not included.

**Figure 3 animals-08-00077-f003:**
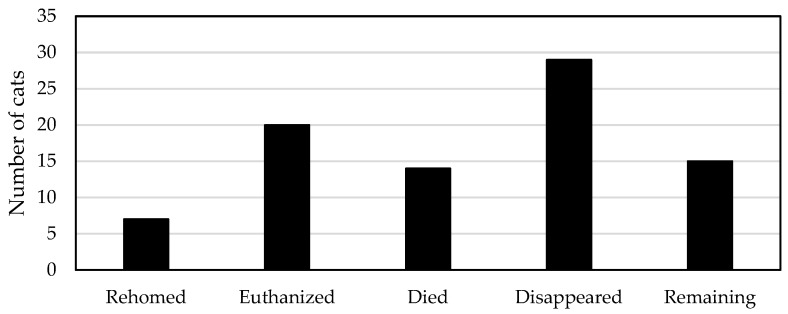
Fates of resident cats (69 original and 16 unsocialized immigrant cats).

**Figure 4 animals-08-00077-f004:**
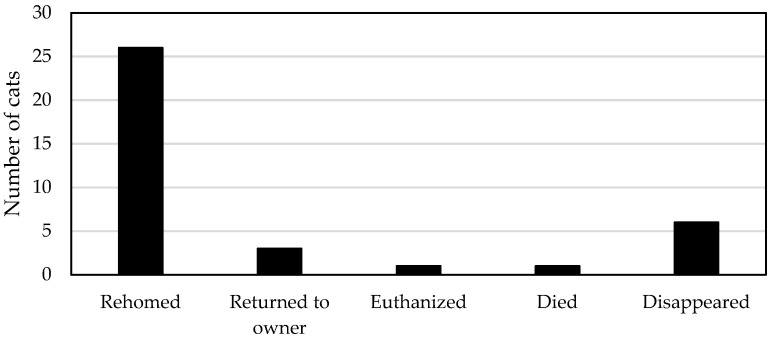
Fates of socialized immigrant adult cats (*n* = 12), solitary kittens (*n* = 6) and kittens born on campus (*n* = 19).

**Table 1 animals-08-00077-t001:** Summary of outcomes for all cats managed in the nine-year program.

	Total	Rehomed	Returned to Owner	Euthanized	Died	Disappeared	Remaining
**RESIDENT CATS**							
Original cats	**69**	7	0	20	12	21	9
Unsocialized immigrant adults	**16**	0	0	0	2	8	6
**Total *n* (%)**	**85**	**7 (8%)**	**0 (0%)**	**20 (24%)**	**14 (16%)**	**29 (34%)**	**15 (18%)**
**NON-RESIDENT CATS**							
Socialized immigrant adults	**12**	7	3	0	0	2	0
Solitary immigrant kittens	**6**	5	0	0	1	0	0
Kittens born on campus	**19**	14	0	1	0	4	0
**Total *n* (%)**	**37**	**26 (70%)**	**3 (8%)**	**1 (3%)**	**1 (3%)**	**6 (16%)**	**0 (0%)**
**OVERALL TOTAL CATS**							
**Total *n* (%)**	**122**	**33 (27%)**	**3 (2%)**	**21 (17%)**	**15 (12%)**	**35 (29%)**	**15 (12%)**

**Table 2 animals-08-00077-t002:** Reasons for euthanasia, categorized by estimated age groups.

Age Group < 5 Years *		Age Group 5–10 Years *		Age Group > 10 Years *	
Acute renal disease	1	Intestinal sarcoma	2	Renal failure	3
Tumorous cancer on chest	1	Saddle thrombus	1	Intestinal sarcoma	1
FIV positive **	2	Feline hypertrophic cardiomyopathy	1	Complications of hyperthyroidism	1
Excessive aggression	1	Aggressive oral squamous cell carcinoma	1	Old age (estimated 20 years)—debilitation	1
Undiagnosed illness	3				
Other	1				
**Total**	**9**	**Total**	**5**	**Total**	**6**

* Age at time of death, based on estimates—see text; ** positive for feline immunodeficiency virus.

**Table 3 animals-08-00077-t003:** Causes of death (other than euthanasia), categorized by estimated age groups.

Age Group <5 Years *		Age Group 5–10 Years *		Age Group >10 Years *	
Hit by car	2	Heart failure **	1	Hit by car	1
Accident (caught in machinery)	1	Severe chronic respiratory disease **	1	Accident (trapped in demolished building)	1
Died during surgery	1	FIV+ ***—unknown cause	1	Elderly—unknown cause	2
Multiple organ lesions **	2				
Unknown cause	1				
**Total**	**7**	**Total**	**3**	**Total**	**4**

* Age at time of death, based on estimates—see text; ** based on necropsy; *** positive for feline immunodeficiency virus.

**Table 4 animals-08-00077-t004:** Comparisons of outcome data from similar TNR studies.

Study (1° Author, Ref)	This Study	Levy [7]	Spehar [18]	Zaunbrecher [14]	Hughes [31]	Centonze [6] *
Duration (years)	9	11	9	3	2	1.5 **
Original cohort (*n*)	69	155	195	41	158	920
Immigration (*n*)	34	NR	NR	6	NR	(103)
Born on site (*n*)	19	NR	NR	NR	NR	(498)
Total managed (*n*)	122	155	195	47	158	920
Remaining (*n*)	15	23	44	36	95	678
Remaining (%)	12	15	23	77	60	74
Rehomed *** (%)	30	47	32	0	21	(26)
Euthanasia (%)	17	11	3	2	15	(0)
Death (%)	12	6	7	11	4	(16)
Disappeared (%)	29	21	34	11	0	(16)
Other (%)	0	0	2	0	0	(0)

NR = not reported. * ambiguous numbers (note that numbers in brackets from the Centonze and Levy paper [6] are unreliable by authors’ admission). ** median—range from two weeks to 15 years. *** includes “returned to owner”.

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
