# Peer review of "Application of a Protocol Based on Trap-Neuter-Return (TNR) to Manage Unowned Urban Cats on an Australian University Campus"

_animals, 2018, doi:10.3390/ani8050077_

Round 1

Reviewer 1 Report

This revision is much improved.  A few items have escaped the authors’ attention, but in general, I have no major comments.

Line 27: Please define or reword “campus users,” which sounds a bit odd.  Presumably these are faculty, staff, and students, but perhaps there are others who “use” the campus.

Line 177: Recommend changing “2007-8” to “2007” to fit the discussion that follows.

Lines 229 and 486: Recommend changing “this paper” to “this study” since it’s the study that is being described in this paper.

Lines 350, 361, and 780: Recommend changing “commencement” to “start” or “beginning” or something less like an academic exercise.

Line 372: Recommend changing “4 female, 9 male” to “four females, nine males” to follow the numbering format style.

Portions of the Discussion section appear to be a continuation of the Results, and this is unusual for a scientific paper.  The Introduction section is rather long.  It is likely that some of the background material in the Introduction section would fit better in the Discussion section so as to streamline the information presented.

Line 756: Recommend deleting the phrase “in this paper” since the reader is already many pages into this paper.

Line 779: Recommend rewording to avoid “this paper” to include an actual Conclusion.  Suggest this: “In summary, the outcomes of a TNR …in Australia include a reduction in the number of resident cats.”

Author Response

This revision is much improved. A few items have escaped the authors’ attention, but in general, I have no major comments.

Comment: Line 27: Please define or reword “campus users,” which sounds a bit odd. Presumably these are faculty, staff, and students, but perhaps there are others who “use” the campus.

Response: This phrase has been changed to: “campus staff and students” (line 27).

Comment: Line 177: Recommend changing “2007-8” to “2007” to fit the discussion that follows.

Response: This phrase has been changed to read: “In January 2008…” (line 179).

Comment: Lines 229 and 486: Recommend changing “this paper” to “this study” since it’s the study that is being described in this paper.

Response: These changes have been made (lines 230 and 489).

Comment: Lines 350, 361, and 780: Recommend changing “commencement” to “start” or “beginning” or something less like an academic exercise.

Response: The word “commencement” has been changed in the text as indicated by the reviewer and at other points in the text where this word was used (lines 218, 238, 243-4, 301, 317, 358, 361, 369, 431, 740, 792, 821, 837).

Comment: Line 372: Recommend changing “4 female, 9 male” to “four females, nine males” to follow the numbering format style.

Response: This change has been made (line 380).

Comment: Portions of the Discussion section appear to be a continuation of the Results, and this is unusual for a scientific paper. The Introduction section is rather long. It is likely that some of the background material in the Introduction section would fit better in the Discussion section so as to streamline the information presented.

Response: No action taken. We have continued to modify the Discussion section in subsequent revisions to consolidate some of this text. Other reviewers have commented positively about the detailed information presented in the Introduction section, and the amount of detail in the Discussion section. Consequently we have not made any further changes.

Comment: Line 756: Recommend deleting the phrase “in this paper” since the reader is already many pages into this paper.

Response: This sentence has been modified to read “This study has a number of limitations…” (line 765).

Comment: Line 779: Recommend rewording to avoid “this paper” to include an actual Conclusion. Suggest this: “In summary, the outcomes of a TNR …in Australia include a reduction in the number of resident cats.”

Response: The first sentence in the Summary section now reads “In summary, a 9-year cat management program conducted on the UNSW Kensington university campus in Sydney, Australia, successfully reduced the campus cat population through a combination of TNR and rehoming.” (line 790-792).

Reviewer 2 Report

This is a good paper which adds to the understanding of TNR.  

In Materials and Methods

Line 216:  Explain "frequent."  Be more specific at what intervals were the audits performed.

Lines 221 & 250:  You introduce the "motion activated cameras" in line 221.  Repetition in line 250 is unnecessary.  

In Results

The caption for Figure 1 states "resident campus male and female cats (including 69 original cats and 16 unsocialized immigrants)" but the graph only includes the 69 resident cats.  Need to change the figure or change the caption.

Line 404:  Consider starting the sentence "By December 2009" rather than "By the December 2009 time point."

Line 476:  Please clarify this sentence.  What is the relevance of the statement "by a concerned student?"  Was the cat injured when the student brought the cat on campus or did the cat get injured after the student brought the cat on campus.

Lines 600-602:  Please clarify that these sentences refer to the one-year Israeli study.

Author Response

This is a good paper which adds to the understanding of TNR.

Comment: In Materials and Methods, Line 216: Explain "frequent." Be more specific at what intervals were the audits performed.

Response: There was no predetermined or regular schedule for audits of the campus cat population, and at various times different areas of the campus received concentrated auditing when concerns arose about local fluctuations in numbers. In general, audits became less frequent in more recent years as the numbers of cats reduced and the colonies settled into their routines. To cover the irregular and often spontaneous nature of the ongoing audit of cat numbers, we have deleted the words “and frequent”, so that this sentence (line 218) now reads “…an ongoing careful audit of campus cats was conducted…”. The use of the word “ongoing” most clearly and reasonably described the nature of the audit process.

Comment: Lines 221 & 250: You introduce the "motion activated cameras" in line 221. Repetition in line 250 is unnecessary.

Response: We have modified both of these sentences. On line 223, the following phrase has been deleted “…particularly newcomers to campus.Line 251 has been reworded to read: “…and this was confirmed where possible by the use of trail cameras.” These changes minimize the repetition while clarifying the use of trail cameras particularly for the identification of immigrant cats.

Comment: In Results, the caption for Figure 1 states "resident campus male and female cats (including 69 original cats and 16 unsocialized immigrants)" but the graph only includes the 69 resident cats. Need to change the figure or change the caption.

Response: No changes made. Figure 1 does in fact present data for both the original residents (n=69) and the 16 unsocialized immigrants. The first set of bars (for Aug-08) adds up to a total of 69 because only original resident cats were on campus at that time. The first immigrant cats did not appear on campus until later in the program, and thus are included in the totals shown in later bars of the graph.

Comment: Line 404: Consider starting the sentence "By December 2009" rather than "By the December 2009 time point."

Response: This change has been made (line 407).

 Comment: Line 476: Please clarify this sentence. What is the relevance of the statement "by a concerned student?" Was the cat injured when the student brought the cat on campus or did the cat get injured after the student brought the cat on campus.

Response: No change made. We feel that the current text (lines 478-479) adequately covers this situation. The kitten died from severe trauma “…sustained prior to being brought on to campus by a concerned student”. The kitten had been found on a roadway by a student, who was unsure how to deal with the distressed animal. The kitten was subsequently brought to campus and handed to the CCC. We took the kitten to the local veterinary practice for examination and care, but it had sustained serious internal injuries consistent with either having been hit by a car or falling from a height, and despite best efforts it survived only for three days.

 Comment: Lines 600-602: Please clarify that these sentences refer to the one-year Israeli study.

Response: To clarify this, we have added the phrase: “In that study,…” (line 605).

Reviewer 3 Report

I would like to congratulate the authors on an excellent paper – it is long (!) but the detail is appreciated to fully comprehend the activities and outcome of this work. Thank you for taking the time to document this work, as TNR efforts so often go unreported.

I was pleased that the contribution of rehoming to the observed reduction in cat populations is noted from the outset. All too often I have seen TNR advocates speak only of the TNR activities and not the complimentary rehoming work – it is the combination of the two, applied on an individual cat basis, that appears most effective in terms of population management and welfare outcomes for cats.

I suggest an addition to the abstract and/or simple summary (one option is to include it on line 41, following the sentence that ends “…rehoming.”) - Include a summary of the application of TNR or rehoming/reuniting by cat numbers or percentage. E.g. of the total 122 cats that experienced intervention (includes original population, kittens born and immigrants); 70% of them were managed using TNR and 30% were rehomed/reunited. This would give the reader a clear picture of the management approach from the outset.

The large immigrant population is common to many TNR sites. Some comment about addressing the source of these cats would be welcomed. TNR does not manage the whole cat population, just those that are unowned, some separate efforts are required to tackle the source of the immigrant population, most usually unwanted owned cats. There is discussion in the limitations section on the need to manage immigrants. Perhaps just one sentence here about the potential benefit of an owned cat desexing programme to reduce immigration?  

The description of the site makes it sound as though the potential for immigration from the owned population was less than would be expected for a ‘normal’ urban site. This is relevant when considering the application of this study’s findings to other locations. Consider including some comment on this in the limitations section. Not essential, only if you agree with my first point about reduced immigration risk in the campus as compared to other sites.

Specific comments by line:

Line 56: “vermin” is quite a loaded term, consider using a more neutral term, e.g. prey.

Line 57: I don’t think the “community” in ‘community cats’ is because they live in colonies; is it not because they are fed by people within the local community? Perhaps better to include “community cats” in the other terms list?

Line 71: I am not sure you need “can become socialised” as some will already be social, for example the cat you describe in the previous sentence, so could just say “unowned urban cats may be adopted…”

Line 90: What does “reasonably” mean in this sentence? Reasonable method of culling or reasonable number of cats culled? Perhaps reword to something like “mainly because the number of cats that can reasonably be culled is an insufficient proportion of the population to exceed the replacement rate…”.

Line 94: Consider changing “human” costs to “emotional and ethical” costs.

Line 96: Add commas for clarity: “There is currently significant interest in Australia, and elsewhere, in reducing this burden…”

Line 128-130: Challenges of enumerating cat populations can be overcome by monitoring the change in density of cats along a standard route or in a standard block – hence enumerating a consistent sample of the population to test effectiveness – you do not need to conduct a census of the whole population to establish whether there has been an impact on population density and welfare. See http://www.acc-d.org/docs/default-source/think-tanks/frc-monitoring-revised-nov-2014.pdf?sfvrsn=2 for guidance.

Line 142-147: An excellent clarification.

Line 169: Consider changing “Subsequently” to “Consequently”.

Line 176: Should be “floundered”.

Section 1.1: I valued this introduction to the situation – it’s possibly an unusual start to an manuscript but I appreciated the scene setting.

Line 239: Add comma for clarity “In other cases, the status is campus cats…”

Line 303: Add commas for clarity “Once the CCC became more familiar with, and confident about, the campus cat population…”

Line 317: For clarity, are you saying the total resident cat population managed using TNR on the campus during the 9-year TNR 317 program was 85 (69+16) cats? I would consider rehoming a form of management, but I think these 85 cats were managed using TNR only? Would be good to clarify this in the text.

Line 376-380: This feels like method to me? Consider moving to section 2.

Line 405 and line 407: “easily” feels like an unnecessary and subjective qualifier for meeting targets – consider deleting these.

Figure 3: Consider turning each bar into a stacked bar colour coded for if these are original or immigrant cats. Only a suggestion, not an essential change.

Line 448: Please clarify if this mortality rate is for adult cats, or if it included kitten mortality? Commonly kitten and adult (> 1 year) mortality are presented separately because they can be so different.

Figure 4: Consider turning each bar into a stacked bar colour coded for the different types of cat. Only a suggestion, not an essential change.

Table 4: This is a very useful table – a great way to visualise the TNR impact spectrum.

Para starting line 539: The difference in euthanasia rate may also have been due to the close audit of cats in your study, where a sick cat could relatively quickly be identified and treated or euthanased following terminal diagnosis. Where less close supervision is given, many cats will instead be found dead. I would view the higher euthanasia rate as evidence of better welfare of your colony, as they were not left to suffer towards the end of their life.

Line 560: Again clarify if this is adult mortality (excludes those cats < 1year) or mortality rate from birth.

Line 585: Add comma for clarity “…but as discussed previously, these comparisons…”

Line 616: citations would be better at the end of the sentence “…kittens, has been reported by others [28,29]”

Line 627-635: In addition to the management of immigrant cats, as you rightly point out. Another interpretation of the challenge presented by immigrant cats is to tackle this population at source: if these are dumped owned cats, then consider the motivations for dumping and whether these can be overcome. TNR is not management of the whole cat population, it targets just those that have already become unowned, there is another usually very large population of owned cats that are a potential source of future unowned cats – these need to be tackled concurrently through different methods.  

Author Response

Comment: I would like to congratulate the authors on an excellent paper – it is long (!) but the detail is appreciated to fully comprehend the activities and outcome of this work. Thank you for taking the time to document this work, as TNR efforts so often go unreported.

I was pleased that the contribution of rehoming to the observed reduction in cat populations is noted from the outset. All too often I have seen TNR advocates speak only of the TNR activities and not the complimentary rehoming work – it is the combination of the two, applied on an individual cat basis, that appears most effective in terms of population management and welfare outcomes for cats.

Response: We thank the reviewer for these generous comments.

Comment: I suggest an addition to the abstract and/or simple summary (one option is to include it on line 41, following the sentence that ends “…rehoming.”) - Include a summary of the application of TNR or rehoming/reuniting by cat numbers or percentage. E.g. of the total 122 cats that experienced intervention (includes original population, kittens born and immigrants); 70% of them were managed using TNR and 30% were rehomed/reunited. This would give the reader a clear picture of the management approach from the outset.

Response: In the Simple Summary, we have added the following: “…managing a further 34 immigrant cats that either joined the resident colony (n=16), were rehomed (n=15), or died/disappeared (n=3).” (lines 24-26)

However, it should be noted that in our program there was not a clear distinction between cats managed with TNR and those that were rehomed – this was not an ‘either/or’ situation. Some cats went through TNR early in the program or before official commencement of the program reported in this paper, and as a result of gradual socialization were later rehomed. The story of Pearl (Appendix B) is a good example. Some socialized but apparently unowned immigrant cats were captured and desexed (TNR) but were almost immediately placed for rehoming rather than returned to campus. Some cats died or disappeared before they could be trapped for desexing. Thus there is no simple way to split the population into those undergoing TNR (only) versus those managed through rehoming (only). We believe that our current wording in the Abstract, together with the detail in the paper and Table 1, adequately summarizes this complexity, and we have not changed the wording here.

 Comment: The large immigrant population is common to many TNR sites. Some comment about addressing the source of these cats would be welcomed. TNR does not manage the whole cat population, just those that are unowned, some separate efforts are required to tackle the source of the immigrant population, most usually unwanted owned cats. There is discussion in the limitations section on the need to manage immigrants. Perhaps just one sentence here about the potential benefit of an owned cat desexing programme to reduce immigration?

Response: We have added the following statement to section 4.1.4 Summary of reasons for reduction in cat population (lines 626-629): “Furthermore, our program did not have the capacity to address management of the source of immigrant cats through community desexing programs for owned cats and promotion of responsible pet ownership in the areas surrounding the campus. The potential impact of such strategies on immigration of owned cats on to the campus is thus unclear.

 Comment: The description of the site makes it sound as though the potential for immigration from the owned population was less than would be expected for a ‘normal’ urban site. This is relevant when considering the application of this study’s findings to other locations. Consider including some comment on this in the limitations section. Not essential, only if you agree with my first point about reduced immigration risk in the campus as compared to other sites.

Response: In the Limitations section (lines 782-783) we already comment that the applicability of our findings outside the limited and defined range of a university campus is unclear. We have added the following: “Immigration might constitute a larger component where colonies are immediately adjacent to houses with free-roaming owned cats, given the limited range of most pet cats [32].” (lines 783-785)

 Specific comments by line:

Comment: Line 56: “vermin” is quite a loaded term, consider using a more neutral term, e.g. prey.

Response: We have replaced the term “vermin” (line 57) with “…prey, mainly introduced mice and rats.”

Comment: Line 57: I don’t think the “community” in ‘community cats’ is because they live in colonies; is it not because they are fed by people within the local community? Perhaps better to include “community cats” in the other terms list?

Response: We have taken the reviewer’s point and have moved this term to the following sentence (line 59).

 Comment: Line 71: I am not sure you need “can become socialised” as some will already be social, for example the cat you describe in the previous sentence, so could just say “unowned urban cats may be adopted…”

Response: We have removed this phrase (line 72).

Comment: Line 90: What does “reasonably” mean in this sentence? Reasonable method of culling or reasonable number of cats culled? Perhaps reword to something like “mainly because the number of cats that can reasonably be culled is an insufficient proportion of the population to exceed the replacement rate…”.

Response: We agree with the reviewer’s suggestion, and this sentence has been reworded (see lines 90-91).

Comment: Line 94: Consider changing “human” costs to “emotional and ethical” costs.

Response: We have made this change (see line 96).

 Comment: Line 96: Add commas for clarity: “There is currently significant interest in Australia, and elsewhere, in reducing this burden…”

Response: Done (line 98).

 Comment: Line 128-130: Challenges of enumerating cat populations can be overcome by monitoring the change in density of cats along a standard route or in a standard block – hence enumerating a consistent sample of the population to test effectiveness – you do not need to conduct a census of the whole population to establish whether there has been an impact on population density and welfare. See http://www.acc-d.org/docs/default-source/think-tanks/frc-monitoring-revised-nov-2014.pdf?sfvrsn=2 for guidance.

Response: We thank the reviewer for this information, and have included the following (see lines 131-132): “…, although methods are available to monitor change in cat density in a subset of the population [27].” A new reference 27, relating to the website suggested by the reviewer, has been added to the reference list.

 Comment: Line 142-147: An excellent clarification.

Response: Thank you.

 Comment: Line 169: Consider changing “Subsequently” to “Consequently”.

Response: This change has been made (line 171).

Comment: Line 176: Should be “floundered”.

Response: Although our use of this term is technically correct, because some readers may not be proficient in English we have replaced the term “foundered” with “…was on the point of failure.” (see line 178).

 Comment: Section 1.1: I valued this introduction to the situation – it’s possibly an unusual start to a manuscript but I appreciated the scene setting.

Response: Thank you.

Comment: Line 239: Add comma for clarity “In other cases, the status of campus cats…”

Response: Done (line 249).

Comment: Line 303: Add commas for clarity “Once the CCC became more familiar with, and confident about, the campus cat population…”

Response: Done (line 309).

Comment: Line 317: For clarity, are you saying the total resident cat population managed using TNR on the campus during the 9-year TNR program was 85 (69+16) cats? I would consider rehoming a form of management, but I think these 85 cats were managed using TNR only? Would be good to clarify this in the text.

Response: It is not quite as straight-forward as this, and is not an ‘either/or’ situation. Some of the original 69 cats were eventually rehomed after earlier having been managed by TNR. This is made clear by reference to Table 1. See also our earlier comment. However, for clarity we have specified the cat numbers here (line 325): “…85 cats (69 originals +16 unsocialized immigrants)

Comment: Line 376-380: This feels like method to me? Consider moving to section 2.

Response: We agree, and have moved this paragraph to the Methods section (lines 269-273), with a slight modification to the header for section 2.2.

 Comment: Line 405 and line 407: “easily” feels like an unnecessary and subjective qualifier for meeting targets – consider deleting these.

Response: The word “easily” has been removed from lines 407 and 409.

 Comment: Figure 3: Consider turning each bar into a stacked bar colour coded for if these are original or immigrant cats. Only a suggestion, not an essential change.

Response: No change made. These data are easily extracted by the reader from Table 1.

 Comment: Line 448: Please clarify if this mortality rate is for adult cats, or if it included kitten mortality? Commonly kitten and adult (> 1 year) mortality are presented separately because they can be so different.

Response: We have added a clarification in line 450 that this estimated mortality rate included both adults and kittens. Kitten deaths were few as most kittens born or found on campus were rapidly removed for rehoming.

 Comment: Figure 4: Consider turning each bar into a stacked bar colour coded for the different types of cat. Only a suggestion, not an essential change.

Response: No change made. These data are easily extracted by the reader from Table 1.

 Comment: Table 4: This is a very useful table – a great way to visualise the TNR impact spectrum.

Response: Thank you.

 Comment: Para starting line 539: The difference in euthanasia rate may also have been due to the close audit of cats in your study, where a sick cat could relatively quickly be identified and treated or euthanased following terminal diagnosis. Where less close supervision is given, many cats will instead be found dead. I would view the higher euthanasia rate as evidence of better welfare of your colony, as they were not left to suffer towards the end of their life.

Response: We thank the reviewer for this interpretation of our data, and have added the following sentence (lines 545-548): “It could also be argued that the close monitoring of our campus population meant that sick cats were identified quickly and euthanased following terminal diagnosis, rather than disappearing or being found dead later due to their illness.

 Comment: Line 560: Again clarify if this is adult mortality (excludes those cats < 1year) or mortality rate from birth.

Response: Done (line 566).

 Comment: Line 585: Add comma for clarity “…but as discussed previously, these comparisons…”

Response: Done (line 590).

Comment: Line 616: citations would be better at the end of the sentence “…kittens, has been reported by others [28,29]”

Response: Done (line 626).

 Comment: Line 627-635: In addition to the management of immigrant cats, as you rightly point out, another interpretation of the challenge presented by immigrant cats is to tackle this population at source: if these are dumped owned cats, then consider the motivations for dumping and whether these can be overcome. TNR is not management of the whole cat population, it targets just those that have already become unowned, there is another usually very large population of owned cats that are a potential source of future unowned cats – these need to be tackled concurrently through different methods.

Response: See our earlier response. We have now addressed this issue in lines 626-629.

Reviewer 4 Report

This is an interesting and in-depth description of an active programme of trap-neuter-release (TNR) conducted over a 9 year period. It is written in a clear and engaging style providing sufficient depth for scientific evaluation and relevance but allowing it to also be accessible to non-technical experts.

The study describes what could be considered a successful programme of humane cat management in a defined locale operated under the constraints of a volunteer / charity funded organisation, with institutional support. To that extent it is very relevant to every day situations and so provides valuable insights about opportunities and limitations for others wishing to implement a similar programme in a limited territory, such as university campuses or other urban environments.

Apart from achieving the desired outcome of reducing cat numbers, the paper describes other outcome measures, such as health status of the cat population and impact on the university human occupants and authorities, demonstrating shifting attitudes to the acceptance of cats within their environment. It addresses concerns often raised by opponents of TNR. and the point made on line 607 is particulalry important in this respect. 

One statement that I might contest is that made starting in line 611. The authors suggest that had the immigrant population of cats not been controlled, this would have resulted in an annual increase in population growth of 4%. Given that the programme was removing cats from the population then it could be argued that this created a 'vacuum' for these immigrant cats to move into. We do not have comparative data to know what would have happened had the population remained stable and the resources available to cats had remained constant. Often cats will migrate when competition in one area becomes too much and resources are greater in another area, which makes that area a more attractive place to be.

A few minor errors were noted:

Line 26: in reduction (insert the word 'a')

Line 496: replace 'an 78%' with 'a 78%'

Author Response

General Comments

Comment: This is an interesting and in-depth description of an active programme of trap-neuter-release (TNR) conducted over a 9 year period. It is written in a clear and engaging style providing sufficient depth for scientific evaluation and relevance but allowing it to also be accessible to non-technical experts.

The study describes what could be considered a successful programme of humane cat management in a defined locale operated under the constraints of a volunteer / charity funded organisation, with institutional support. To that extent it is very relevant to every day situations and so provides valuable insights about opportunities and limitations for others wishing to implement a similar programme in a limited territory, such as university campuses or other urban environments.

Apart from achieving the desired outcome of reducing cat numbers, the paper describes other outcome measures, such as health status of the cat population and impact on the university human occupants and authorities, demonstrating shifting attitudes to the acceptance of cats within their environment. It addresses concerns often raised by opponents of TNR, and the point made on line 607 is particularly important in this respect. 

Response: We thank the reviewer for these generous comments.

 Comment: One statement that I might contest is that made starting in line 611. The authors suggest that had the immigrant population of cats not been controlled, this would have resulted in an annual increase in population growth of 4%. Given that the programme was removing cats from the population then it could be argued that this created a 'vacuum' for these immigrant cats to move into. We do not have comparative data to know what would have happened had the population remained stable and the resources available to cats had remained constant. Often cats will migrate when competition in one area becomes too much and resources are greater in another area, which makes that area a more attractive place to be.

Response: We thank the reviewer for this comment.  We have added the following sentence into our discussion of the impact of immigrant cats at lines 619-623: “At the same time, it must be acknowledged that our program reduced overall cat numbers (see Figure 1 and Appendix A), potentially creating the circumstances in which a “vacuum effect” may have encouraged the influx of immigrant cats. We cannot know what would have happened had the resident population remained stable and resources to support the cat population had remained constant. Nevertheless, …

Comment: A few minor errors were noted:

Line 26: in reduction (insert the word 'a')

Line 496: replace 'an 78%' with 'a 78%'

Response: these minor corrections have been done (line 26, line 498).